# Lipoprotein concentrations over time in the intensive care unit COVID-19 patients: Results from the ApoCOVID study

Sébastien Tanaka[1,2]*, Christian De Tymowski[1,3,4], Maksud Assadi[1,4], Nathalie Zappella[1], Sylvain Jean-Baptiste[1], Tiphaine Robert[5], Katell Peoc'h[3,4,5], Brice Lortat-Jacob[1], Lauriane Fontaine[1], Donia Bouzid[4,6,7], Alexy Tran-Dinh[1,8], Parvine Tashk[1], Olivier Meilhac[2,9], Philippe Montravers[1,4,10]

1 Assistance Publique—Hôpitaux de Paris (AP-HP), Department of Anesthesiology and Critical Care Medicine, Bichat-Claude Bernard Hospital, Paris, France, 2 Réunion Island University, French Institute of Health and Medical Research (INSERM), U1188 Diabetes atherothrombosis Réunion Indian Ocean (DéTROI), CYROI Plateform, Saint-Denis de La Réunion, Réunion, France, 3 French Institute of Health and Medical Research (INSERM) U1149, Center for Research on Inflammation, Paris, France, 4 University of Paris, UFR Denis Diderot, Paris, France, 5 Assistance Publique—Hôpitaux de Paris (AP-HP), Biochemistry Department, Bichat-Claude Bernard Hospital, Paris, France, 6 INSERM U1137 IAME, Paris, France, 7 Assistance Publique—Hôpitaux de Paris (AP-HP), Emergency Department, Bichat-Claude Bernard Hospital, Paris, France, 8 French Institute of Health and Medical Research (INSERM) U1148, Laboratory for Vascular Translational Science, Paris, France, 9 Réunion Island University-affiliated Hospital, Réunion, France, 10 French Institute of Health and Medical Research (INSERM) U1152, Physiopathology and Epidemiology of Respiratory Diseases, Paris, France

☯ These authors contributed equally to this work.
* sebastien.tanaka@aphp.fr

**Data Availability Statement:** All relevant data are within the manuscript and its Supporting Information files.

## Abstract

### Introduction

Severe acute respiratory syndrome coronavirus2 has caused a global pandemic of coronavirus disease 2019 (COVID-19). High-density lipoproteins (HDLs), particles chiefly known for their reverse cholesterol transport function, also display pleiotropic properties, including anti-inflammatory or antioxidant functions. HDLs and low-density lipoproteins (LDLs) can neutralize lipopolysaccharides and increase bacterial clearance. HDL cholesterol (HDL-C) and LDL cholesterol (LDL-C) decrease during bacterial sepsis, and an association has been reported between low lipoprotein levels and poor patient outcomes. The goal of this study was to characterize the lipoprotein profiles of severe ICU patients hospitalized for COVID-19 pneumonia and to assess their changes during bacterial ventilator-associated pneumonia (VAP) superinfection.

### Methods

A prospective study was conducted in a university hospital ICU. All consecutive patients admitted for COVID-19 pneumonia were included. Lipoprotein levels were assessed at admission and daily thereafter. The assessed outcomes were survival at 28 days and the incidence of VAP.

**Funding:** The author(s) received no specific funding for this work.

**Competing interests:** The authors have declared that no competing interests exist.

## Results

A total of 48 patients were included. Upon admission, lipoprotein concentrations were low, typically under the reference values ([HDL-C] = 0.7[0.5–0.9] mmol/L; [LDL-C] = 1.8[1.3–2.3] mmol/L). A statistically significant increase in HDL-C and LDL-C over time during the ICU stay was found. There was no relationship between HDL-C and LDL-C concentrations and mortality on day 28 (log-rank $p = 0.554$ and $p = 0.083$, respectively). A comparison of alive and dead patients on day 28 did not reveal any differences in HDL-C and LDL-C concentrations over time. Bacterial VAP was frequent (64%). An association was observed between HDL-C and LDL-C concentrations on the day of the first VAP diagnosis and mortality ([HDL-C] = 0.6[0.5–0.9] mmol/L in survivors vs. [HDL-C] = 0.5[0.3–0.6] mmol/L in nonsurvivors, $p = 0.036$; [LDL-C] = 2.2[1.9–3.0] mmol/L in survivors vs. [LDL-C] = 1.3[0.9–2.0] mmol/L in nonsurvivors, $p = 0.006$).

## Conclusion

HDL-C and LDL-C concentrations upon ICU admission are low in severe COVID-19 pneumonia patients but are not associated with poor outcomes. However, low lipoprotein concentrations in the case of bacterial superinfection during ICU hospitalization are associated with mortality, which reinforces the potential role of these particles during bacterial sepsis.

## Introduction

In late 2019, the novel coronavirus severe acute respiratory syndrome coronavirus 2 (SARS-CoV-2) was identified as the cause of COVID-19 in Hubei Province, China [1]. COVID-19 has since become a pandemic, and by May 2020 more than 4.1 million confirmed cases had been reported, with more than 280,000 attributable deaths worldwide [2]. Five to 20% of patients hospitalized with COVID-19 are admitted to the intensive care unit (ICU), with a mortality rate ranging from 25% to 60% [3–5]. At present, there is no effective specific treatment for COVID-19 described in the surviving sepsis campaign recommendations [6].

Lipoprotein particles are defined by their composition (lipids and proteins) and are classified according to their density (from very low to high density). They have the capacity to transport hydrophobic lipids (e.g., cholesterol) in a hydrophilic environment (plasma), but they also exert numerous pleiotropic properties [7]. High-density lipoproteins (HDLs), responsible for reverse cholesterol transport (RCT), display endothelioprotective functions, including anti-inflammatory, anti-apoptotic, and antioxidant effects; they can also bind and neutralize lipopolysaccharides (LPS), enhancing LPS clearance [8–11]. Numerous clinical studies have reported a marked decrease in the concentration of HDL cholesterol (HDL-C) during bacterial sepsis [12–19], and this decrease is currently associated with poor outcomes [17–20]. Experimental studies assessing both reconstituted HDL (rHDL) and ApoA-I mimetic peptide perfusion in animal models of septic shock have been performed, demonstrating protective effects against morbidity and mortality [21–23]. Low-density lipoproteins (LDLs) also neutralize LPS [24, 25], and observational studies have reported that LDL-cholesterol (LDL-C) concentrations could decrease by 30% in inflammatory states such as sepsis [19, 26]. Walley et al. also showed that low LDL-C concentrations were associated with a poor prognosis in septic patients [27].

The links between viral infection and lipoproteins are less clear than the association between bacterial infection and lipoproteins [7]. Lipid profiles are reportedly altered during

some viral infections, such as dengue, in which Total cholesterol (TC) and lipoprotein levels are frequently low; specifically, a low LDL-C concentration is correlated with severity [28, 29]. In a recent study involving 21 mixed ICU and non-ICU COVID-19 patients, LDL-C concentrations were significantly decreased upon admission and were inversely correlated with disease severity [30]. Most patients with COVID-19 in the ICU develop bacterial ventilator-associated pneumonia (VAP) [31], suggesting that both COVID-19 and bacterial infection may influence the lipid profile.

Here, we aimed to characterize the lipid profiles of patients with severe COVID-19 pneumonia upon their ICU admission and during hospitalization, with a particular focus on changes in HDL-C and LDL-C concentrations. We also assessed lipid profile changes in these patients during bacterial VAP superinfection.

## Materials and methods

This was a monocentric study conducted in the surgical ICU of Bichat Claude-Bernard University Hospital, Paris, France. Patients admitted between March 18, 2020 and April 13, 2020 for acute respiratory distress syndrome (ARDS) due to COVID-19 pneumonia were consecutively and prospectively included in a database, and their medical charts were reviewed retrospectively. The French Society of Anesthesiology and Critical Care Medicine Research Ethics Board approved this study and waived the need for consent (ApoCOVID study, IRB 00010254-2020-082).

Patient demographic information, Simplified Acute Physiology Score II (SAPSII), Sepsis-related Organ Failure Assessment (SOFA) severity scores and clinical data were collected. Inflammatory parameters (leukocytes, lymphocytes, C-reactive protein and procalcitonin) were also assessed. Mortality at 28 days, duration of mechanical ventilation, number of days alive without mechanical ventilation at day 28, length of stay in the ICU and in the hospital, renal replacement therapy, vasopressor use, need for extracorporeal membrane oxygenation (ECMO), and tracheostomy were collected. The number of prone positioning procedures and instances of ventilator-associated pneumonia were also collected.

Plasma concentrations of TC, HDL-C, LDL-C, and triglycerides (TG) were measured upon ICU admission and then daily in the Biochemistry Laboratory of Bichat Claude-Bernard Hospital. Owing to problems in the supply of anesthetic drugs during the COVID-19 epidemic, we had to administer propofol to all patients over a long period of time, which is why we measured daily lipid levels. TC, HDL-C, LDL-C and triglyceride concentrations were determined by routine enzymatic assays (CHOL, HDL-C, LDL-C and TRIG methods, Dimension VISTA® System, Siemens Healthineers™). The reference values for these assays were as follows: HDL-C: > 1.40 mmol/L; total cholesterol (TC): $4.40 < N < 5.2$ mmol/L; and triglycerides: $0.50 < N < 1.7$ mmol/L. According to the French National Authority for Health 2017 and the European Society of Cardiology 2016 recommendations, LDL-C concentration targets have been established depending on vascular risk factors [32]. According to these recommendations, LDL-C <3.0 mmol/L is advised in low- to moderate-risk patients, and LDL-C <2.6 mmol/L is recommended in high-cardiovascular risk patients.

Ventilator-associated pneumonia (VAP) diagnosis was based on the Infectious Diseases Society of America and the American Thoracic Society guidelines [33].

Statistical analysis: Continuous variables were expressed as medians with interquartile ranges (IQRs) and were compared using the Mann-Whitney U test. Categorical variables were expressed as counts and percentages and were compared using Fisher's exact test or the chi-square test, as appropriate. The threshold defining the lower quartile was determined to have 25% of the overall population in that quartile. Survival was estimated by Kaplan-Meier analysis

and compared by the log-rank test. Correlations were assessed by Spearman's rank-order correlation. A mixed model for repeated measures was performed to compare the lipoproteins' evolution over time in the overall population and according to 28-day mortality. All statistical analyses were performed using SPSS software, version 21 (IBM, Armonk, NY, USA). P-values < 0.05 were considered statistically significant.

## Results

### a. Population

From March 18, 2020 to April 13, 2020, 67 patients were hospitalized for COVID-19 pneumonia in our ICU. Because of a lack of lipid profile data at admission for 19 patients, 48 patients were finally included in the study. Twenty-nine patients were directly admitted into the ICU, while 19 patients had a short stay (1 [0–3] days) in another ward before ICU admission. The general characteristics and outcomes of the patients are presented in Table 1. Comparisons between survivors (n = 32, 67%) and nonsurvivors (n = 16, 33%) at day 28 are presented in Table 1. At ICU admission, no patient had a bacterial coinfection associated with COVID-19 pneumonia.

### b. Lipid concentrations at admission and kinetics over time

Upon ICU admission, the TC, TG, HDL-C and LDL-C concentrations were 3.2 [2.5–4.0] mmol/L, 2.0 [1.6–2.9] mmol/L, 0.7 [0.5–0.9] mmol/L and 1.8 [1.3–2.3] mmol/L, respectively. All results except the TG concentrations were below the abovementioned reference values (see Materials and Methods section).

Fig 1 exhibits the kinetics of the TC, TG, HDL-C and LDL-C concentrations throughout the ICU stay. Except for the TG concentration, which remained within the normal reference value range, we observed a statistically significant increase in TC, HDL-C and LDL-C during the study period (p<0.001, p = 0.024 and p<0.001, respectively); these values returned to normal in the survivors.

### c. Relationship between lipid concentrations and COVID-19-specific therapies

We found no differences in the lipid concentrations upon admission or over time between patients with and without lopinavir/ritonavir treatment (see S1 Fig). In addition, corticosteroid therapy did not significantly alter the lipid concentrations upon admission or over time (see S2 Fig).

### d. Relationship between lipid concentrations and mortality

Upon ICU admission, there were no difference in lipid concentrations between survivors and nonsurvivors (TC: 3.3 [2.5–4.0] mmol/L in survivors vs. 3.0 [2.2–4.1] mmol/L in nonsurvivors, p = 0.43; TG: 2.0 [1.5–2.9] mmol/L in survivors vs. 2.0 [1.7–3.0] mmol/L in nonsurvivors; p = 0.827; HDL-C: 0.7 [0.5–1.1] mmol/L in survivors vs. 0.7 [0.4–0.8] mmol/L in nonsurvivors, p = 0.459; LDL-C: 1.9 [1.4–2.4] in survivors vs. 1.5 [1.1–2.2] mmol/L in nonsurvivors, p = 0.262). In addition, changes in the TC, TG, HDL-C and LDL-C concentrations over time during the first six days following ICU admission did not allow us to separate patients as alive or dead at day 28 (Fig 2).

Fig 3 shows the mortality at day 28 according to the lipid profile upon ICU admission as estimated by the Kaplan-Meier analysis and compared by the log-rank test. No relationship was found between patients with TC, TG, HDL-C and LDL-C concentrations in their

**Table 1. General characteristics and outcome of the patients.**

| Characteristics | Overall population (n = 48) | Alive at day 28 (n = 32; 67%) | Dead at day 28 (n = 16; 33%) | p Value |
|---|---|---|---|---|
| Age, years, median [IQR] | 57 [46–64] | 55 [45–62] | 59 [50–67] | 0.283 |
| Male sex, n (%) | 31 (65) | 21 (65) | 10 (63) | 0.831 |
| BMI, kg/m$^2$, median [IQR] | 27.9 [25–31] | 27 [24–29.6] | 29.7 [26–35.5] | 0.135 |
| **Presence of comorbidities / medications** | | | | |
| High blood pressure, n (%) | 24 (50) | 14 (44) | 10 (63) | 0.221 |
| ACEI or ARB use, n (%) | 16 (33) | 9 (28) | 7 (44) | 0.279 |
| Diabetes mellitus, n (%) | 17 (35) | 8 (25) | 9 (56) | 0.033 |
| Statin use, n (%) | 13 (27) | 7 (22) | 6 (37) | 0.310 |
| **Timing of hospitalization** | | | | |
| Between first symptoms and hospitalization (days) | 6 [3–7] | 6 [4–8] | 5 [2–7] | 0.235 |
| Between hospitalization and ICU admission (days) | 1 [0–3] | 2 [0–3] | 0 [0–4] | 0.530 |
| **Severity scores at admission** | | | | |
| SAPSII, median [IQR] | 43 [33–53] | 40 [31–51] | 50 [44–28] | 0.006 |
| SOFA, median [IQR] | 5 [4–7] | 5 [4–7] | 6 [4–7] | 0.241 |
| **Inflammatory parameters at admission** | | | | |
| Leukocyte count (G/L) | 8.7 [6.5–12.4] | 9.7 [7–14] | 7.6 [6.4–11.7] | 0.155 |
| Lymphocyte count (G/L) | 0.8 [0.6–1.3] | 0.9 [0.6–1.3] | 0.7 [0.4–1.3] | 0.323 |
| Procalcitonin (μg/L) | 0.8 [0.3–3.3] | 0.8 [0.4–2.7] | 0.6 [0.3–3.6] | 0.850 |
| C-reactive protein (mg/L) | 136 [97–219] | 136 [102–209] | 145 [86–246] | 0.786 |
| **Treatments during ICU stay** | | | | |
| Norepinephrine, n (%) | 32 (67) | 19 (60) | 13 (81) | 0.130 |
| Mechanical ventilation, n (%) | 46 (96) | 30 (94) | 16 (100) | 0.546 |
| Length of mechanical ventilation, median [IQR] | 12 [7–25] | 20 [7–28] | 7 [6–9] | 0.007 |
| Prone positioning, n (%) | 33 (69) | 21 (66) | 12 (75) | 0.509 |
| Tracheostomy, n (%) | 11 (23) | 10 (31) | 1 (6) | 0.073 |
| ECMO, n (%) | 10 (20) | 3 (9) | 7 (44) | 0.010 |
| RRT, n (%) | 13 (27) | 9 (28) | 4 (25) | 1 |
| **COVID-specific treatments** | | | | |
| Lopinavir/ritonavir, n (%) | 6 (12) | 5 (16) | 1 (6) | 0.648 |
| Hydroxychloroquine, n (%) | 3 (6) | 2 (6) | 1 (6) | 1 |
| Corticosteroids, n (%) | 12 (25) | 10 (31) | 2 (12) | 0.289 |
| **Outcome** | | | | |
| ICU LOS, median [IQR] | 12 [7–27] | 22 [10–33] | 7 [6–9] | 0.002 |
| Hospital LOS, median [IQR] | 20 [7–31] | 28 [19–38] | 7 [6–09] | <0.001 |
| Mortality at day 28, n (%) | 16 (33) | - | - | - |

BMI, body mass index; ECMO, extracorporeal membrane oxygenation; LOS, length of stay; RRT, renal replacement therapy; SAPS II, simplified acute physiology score II; ACEI, angiotensin-converting-enzyme inhibitor; ARB, angiotensin II receptor blocker.

respective lower quartile and mortality at day 28 (log-rank p = 0.092, p = 0.611, p = 0.554 and p = 0.083, respectively).

## e. Relationship between the lipid concentrations and patient outcomes

The data in Table 2 show the relationships between TC, TG, HDL-C and LDL-C concentrations upon ICU admission and the ICU outcomes (ventilator-associated pneumonia, renal replacement therapy, need for norepinephrine). No statistically significant links were found between TC, TG, HDL-C or LDL-C at admission and these ICU outcomes.

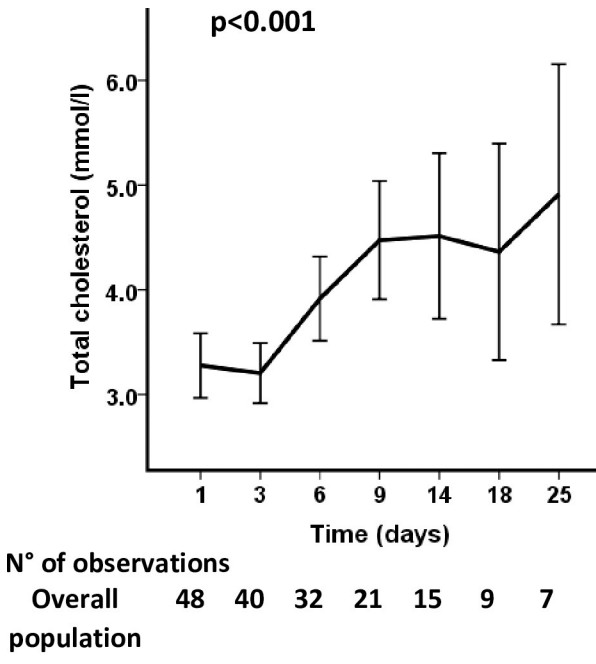

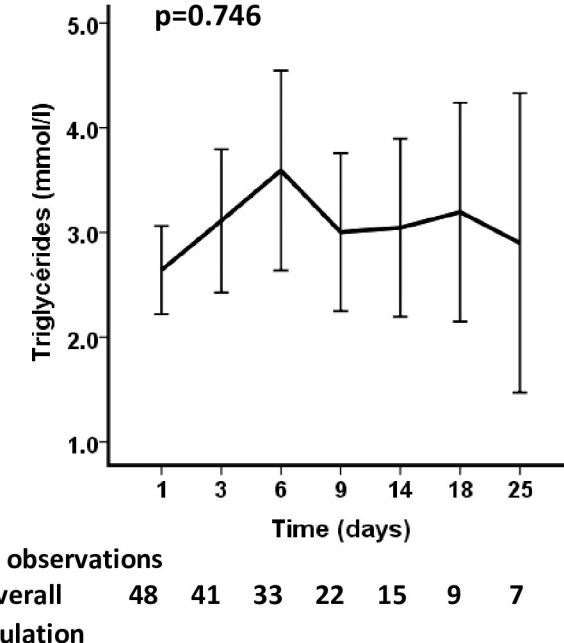

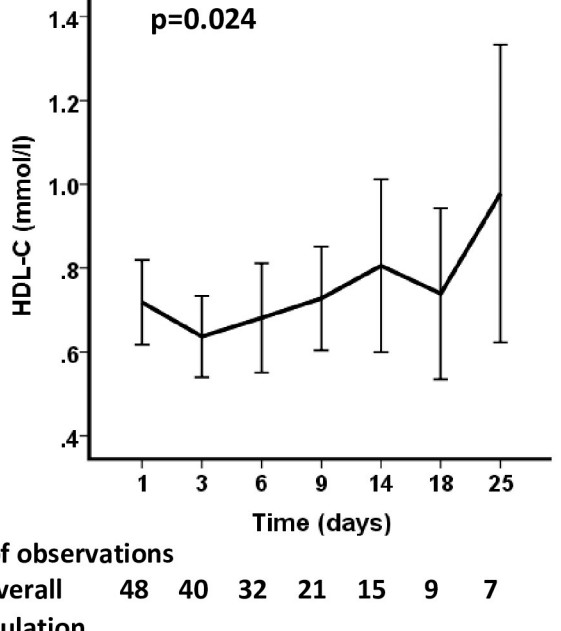

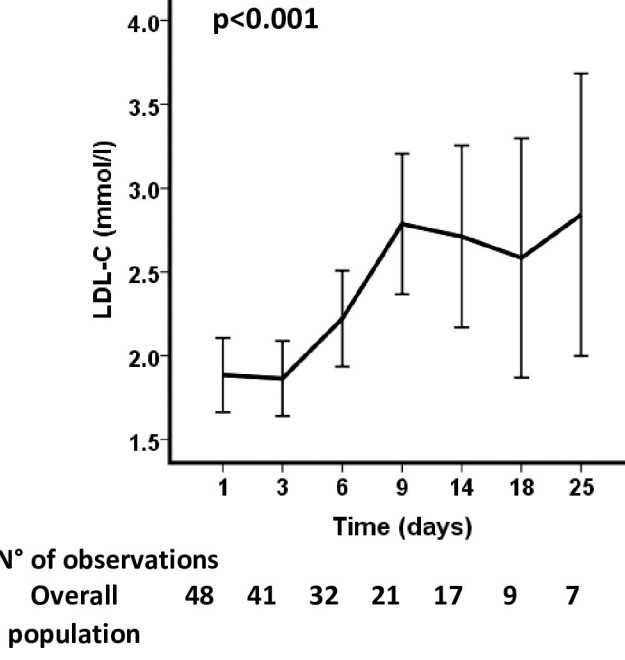

**Fig 1. Total cholesterol, triglycerides, HDL-C and LDL-C kinetics during the ICU stay.**

There were no correlations between TC, TG, HDL-C or LDL-C concentrations upon ICU admission and length of ICU stay, length of hospital stay, or the number of days alive without mechanical ventilation (S1 Table).

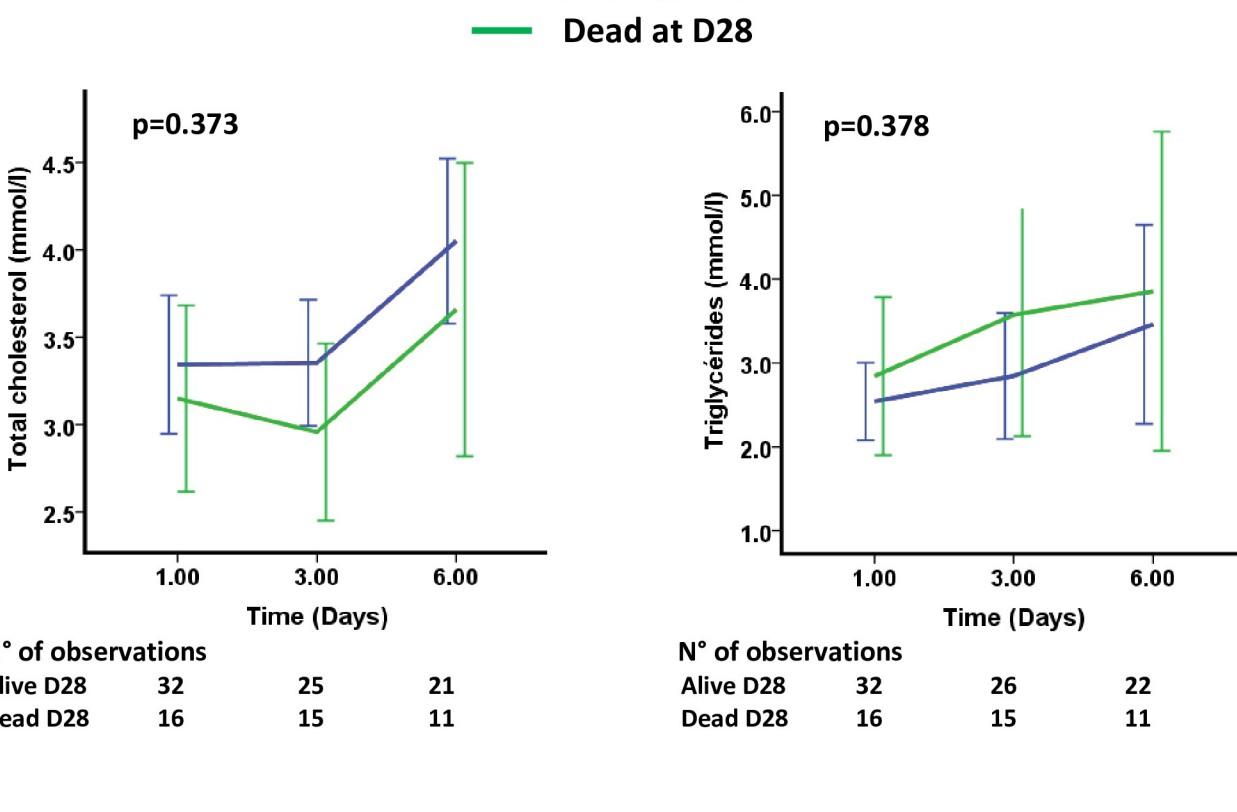

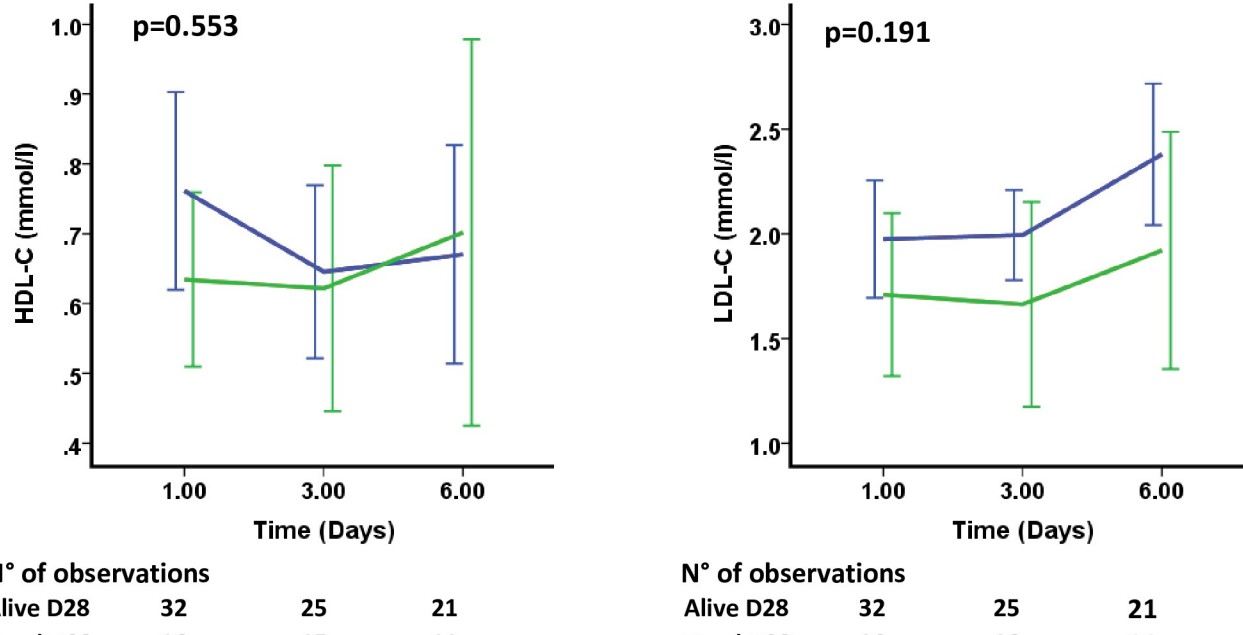

**Fig 2. Kinetics of total cholesterol, triglycerides, HDL-C and LDL-C concentrations over the first six days according to their status outcome (dead or alive at day 28).**

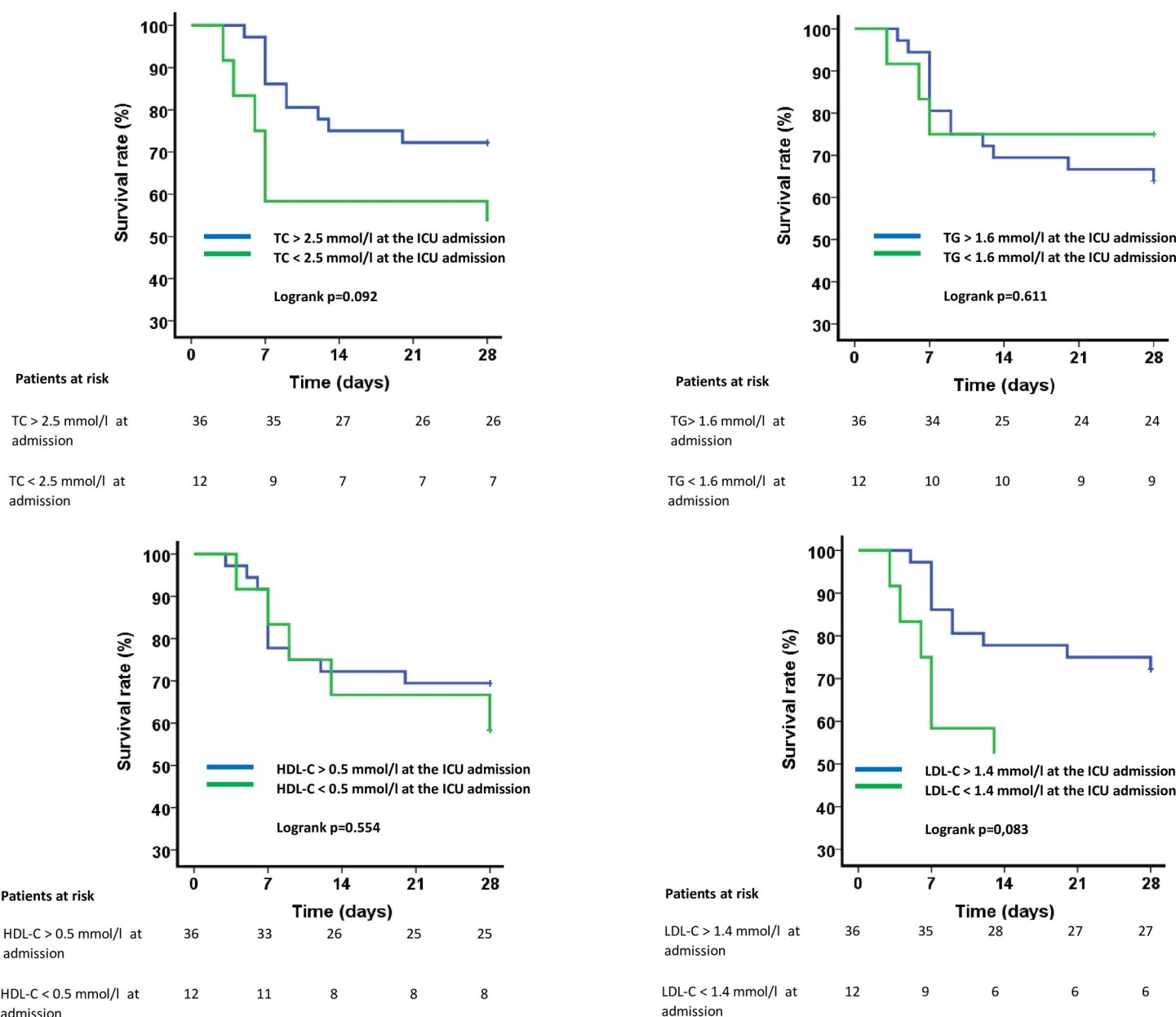

**Fig 3. Mortality at day 28 according to the lipid profile upon ICU admission.** Survival analyses were estimated by Kaplan-Meier analysis and compared using the log-rank test.

## f. Relationship between lipid concentrations and mortality in the subgroup of patients with bacterial ventilated-associated pneumonia (VAP)

To better characterize the relationship between the TC, TG, HDL-C and LDL-C concentrations and the outcomes, we focused on COVID-19 patients with bacterial VAP during their stay in the ICU (n = 29, 64%). The median period of time before the onset of VAP was 7 [3–9] days, and the median number of VAP episodes for each patient was 2 [1–3]. Concentrations of TC, TG, HDL-C, and LDL-C were measured on the day of VAP diagnosis. A statistically significant association was found between HDL-C and LDL-C concentrations on the day of the

**Table 2. Relationship between lipid concentrations at ICU admission and outcome variables.**

| Lipid concentrations at ICU admission | Overall population (n = 48) | VAP (n = 29; 64%) | No VAP (n = 19; 36%) | p Value |
|---|---|---|---|---|
| Total cholesterol, mmol/L, median [IQR] | 3.2 [2.5–4.0] | 3.1 [2.5–3.9] | 3.5 [2.5–4.2] | 0.349 |
| Triglycerides, mmol/L, median [IQR] | 2.0 [1.6–2.9] | 2.1 [1.8–3.0] | 1.9 [1.4–2.6] | 0.260 |
| HDL-C, mmol/L, median [IQR] | 0.7 [0.5–0.9] | 0.7 [0.4–0.8] | 0.8 [0.6–1.1] | 0.077 |
| LDL-C, mmol/L, median [IQR] | 1.8 [1.3–2.3] | 1.7 [1.1–2.3] | 2.0 [1.4–2.6] | 0.266 |
| **Lipid concentrations at ICU admission** | **Overall population (n = 48)** | **RRT (n = 13; 27%)** | **No RRT (n = 35; 73%)** | **p Value** |
| Total cholesterol, mmol/L, median [IQR] | 3.2 [2.5–4.0] | 3.4 [2.5–4.0] | 2.8 [2.4–4.0] | 0.609 |
| Triglycerides, mmol/L, median [IQR] | 2.0 [1.6–2.9] | 1.9 [1.5–2.8] | 2.1 [1.8–2.9] | 0.372 |
| HDL-C, mmol/L, median [IQR] | 0.7 [0.5–0.9] | 0.8 [0.5–1.0] | 0.6 [0.4–0.8] | 0.164 |
| LDL-C, mmol/L, median [IQR] | 1.8 [1.3–2.3] | 1.9 [1.4–2.4] | 1.5 [1.1–2.2] | 0.439 |
| **Lipid concentrations at ICU admission** | **Overall population (n = 48)** | **NOR (n = 32; 67%)** | **No NOR (n = 16; 33%)** | **p Value** |
| Total cholesterol, mmol/L, median [IQR] | 3.2 [2.5–4.0] | 3.4 [2.5–3.9] | 3.1 [2.5–4.1] | 0.864 |
| Triglycerides, mmol/L, median [IQR] | 2.0 [1.6–2.9] | 2.2 [1.7–2.8] | 1.9 [1.6–2.9] | 0.743 |
| HDL-C, mmol/L, median [IQR] | 0.7 [0.5–0.9] | 0.7 [0.6–1.1] | 0.7 [0.5–0.9] | 0.458 |
| LDL-C, mmol/L, median [IQR] | 1.8 [1.3–2.3] | 1.8 [1.4–2.3] | 1.7 [1.2–2.4] | 0.927 |

Variables: ventilator-associated pneumonia, renal replacement therapy, need for norepinephrine and need for extracorporeal membrane oxygenation. Continuous variables are expressed as the median and interquartile range (IQR). VAP: ventilator-associated pneumonia; RRT: renal replacement therapy; NOR: norepinephrine.

first VAP diagnosis and mortality on day 28 ([HDL-C] = 0.6 [0.5–0.9] mmol/L in patients alive on day 28 vs. [HDL-C] = 0.5 [0.3–0.6] mmol/L in patients dead on day 28, p = 0.036; [LDL-C] = 2.2 [1.9–3.0] mmol/L in patients alive on day 28 vs. [LDL-C] = 1.3 [0.9–2.0] mmol/L in patients dead on day 28, p = 0.006). No statistically significant association was found between the TC or TG concentrations and mortality at day 28. Box plots are presented in Fig 4.

## Discussion

In this cohort of patients who were hospitalized in the ICU for COVID-19 pneumonia, we observed that lipoprotein concentrations upon ICU admission were low but were not significantly related to the patient's outcomes. In contrast, in the subgroup of patients who developed bacterial VAP, a strong association was found between mortality and the concentrations of HDL-C and LDL-C at the time of VAP diagnosis.

The low concentrations of HDL-C and LDL-C upon admission reported here are consistent with previous results obtained in two studies conducted by the same Chinese team in a cohort of heterogeneous patients hospitalized for COVID-19 [30, 34]. In these studies, the authors compared the lipid profiles of COVID-19 patients and healthy subjects and reported differences between the two groups. However, it should be noted that the severity scores of these two populations (COVID-19 and healthy volunteers) were not compared. The study involving 597 COVID-19 patients stratified into mild (n = 394), severe (n = 171) and critical (n = 32) cases showed that LDL-C and TG concentrations were significantly different between the groups, with lower concentrations in the most severe patients [34]. Moreover, low levels of HDL-C and LDL-C in our population are in accordance with another study involving 114 COVID-19 patients [35]. In this study, TC, HDL-C and LDL-C concentrations in COVID-19 patients were low but were also significantly lower than the values of age-matched healthy control patients.

During bacterial sepsis, a decrease in lipoproteins during the acute phase is well documented (in particular for HDL-C); however, the mechanisms underlying this decrease are

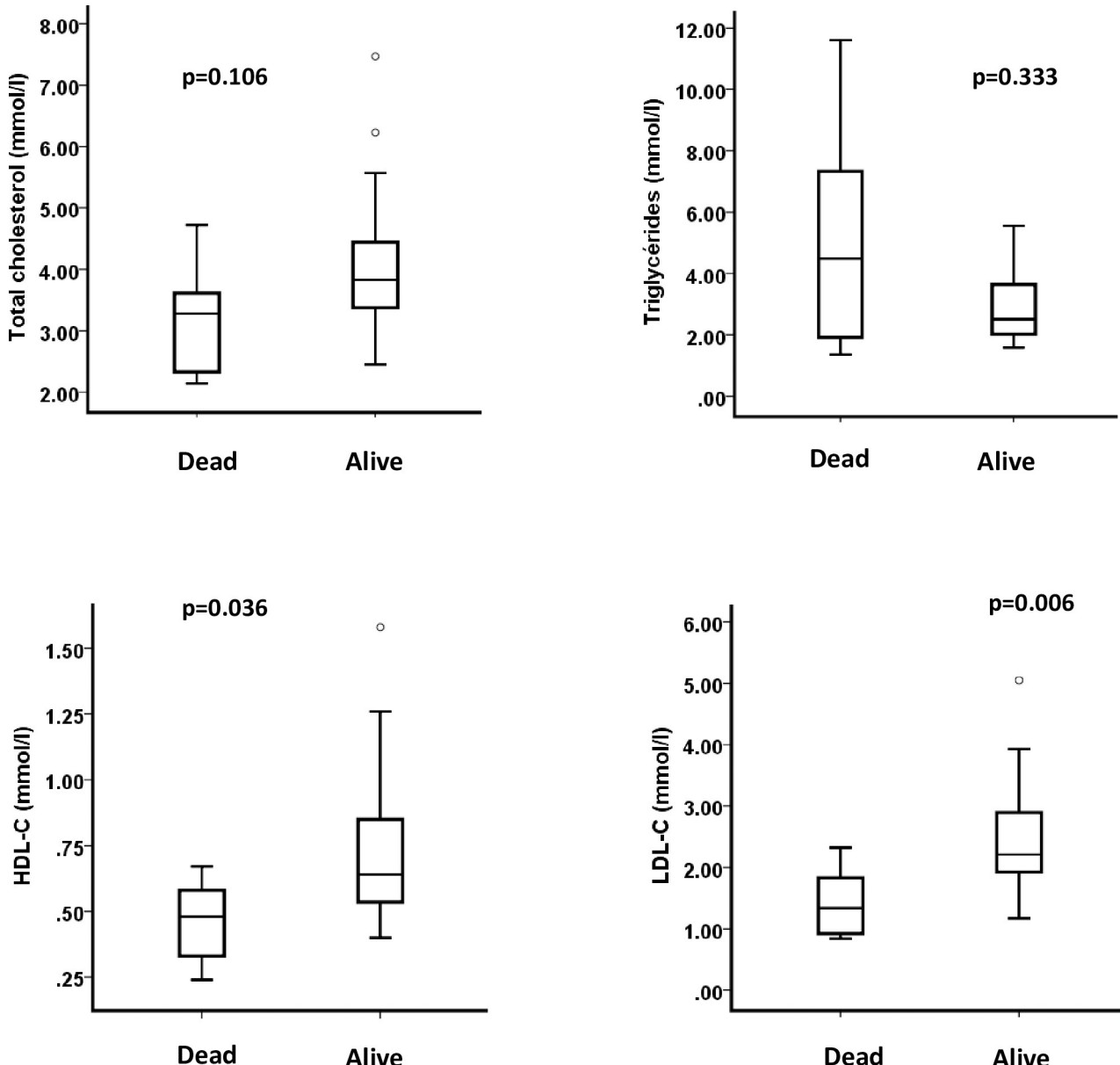

**Fig 4. Relationship between total cholesterol, triglycerides, HDL-C and LDL-C concentrations on the day of the first VAP diagnosis and mortality on day 28.**

poorly described [11]. Several hypotheses have been proposed to explain this HDL-C concentration decrease such as the consumption of HDL particles, hemodilution, capillary leakage, decreased HDL synthesis by the liver (particularly in cases of associated liver dysfunction) or increased HDL clearance following the upregulation of scavenger receptor class B type 1 (SRB1) expression [20]. During COVID-19, similar potential mechanisms might be present. A pulmonary histological study involving COVID-19 patients documented increased vascular permeability [36]. The considerable inflammation described during COVID-19 ARDS could potentially participate in the lipoprotein concentration decrease [37, 38]. Additionally, the direct action of the virus on lipoproteins should not be ruled out, but this hypothesis requires further investigation.

Notably, we have shown a gradual increase over time in both the HDL-C and LDL-C concentrations. These increases over time could be related to patient improvement, the restoration of normovolemia, a reduced inflammatory state, decreased capillary leakage or increased hepatic HDL synthesis [7, 11]. The findings described herein of decreased lipoprotein levels at admission followed by increases over time are in accordance with the studies by Fan et al. and Hu et al. [30, 39].

In our cohort of 48 severe COVID-19 patients hospitalized in our ICU, we found no statistically significant link between HDL-C or LDL-C concentrations and patient outcomes. These results are not in accordance with the results of the study by Fan et al., which showed that a low LDL-C concentration was a potential predictor of poor prognosis in a cohort of 21 heterogeneous ICU and non-ICU COVID-19 patients [30]. Interestingly, the studies by Hu et al. and Wei et al. have specifically shown that low HDL-C levels were associated with severe COVID-19 disease [34, 39]. However, the heterogeneity reported in these two studies, which included mild, severe and critical patients and ICU and non-ICU patients, does not allow for direct comparisons with our specific severe COVID-19 ICU cohort [34, 39].

In contrast to studies involving septic patients due to bacteria, in which HDL-C and LDL-C concentrations are correlated with the outcome [11, 17, 19, 22, 27], data concerning viral infections appear to be more controversial. While altered lipid profiles have been described in some viral pathologies, such as HIV (frequently associated with increased TC and LDL-C and decreased HDL-C) and hepatitis B virus infections, direct links to patient outcomes remain uncertain [40–42]. We hypothesize that during sepsis in the context of bacterial infections, lipoprotein concentrations seem to be a determinant for the patient's outcome. Pleiotropic properties of lipoproteins and, in particular, of HDL particles, such as anti-inflammatory, anti-apoptotic or antioxidant effects, could play an important role. The ability of HDL to bind and inactivate LPS and lipoteichoic acid, leading to a potential increase in bacterial clearance, appears to be an essential function that could support the strong association between the HDL concentrations and patient outcomes [11, 43, 44]. The lack of a statistically significant association in our study focusing on viral pathology might be related to the specific action of HDL particles on bacteria. The relationship between mortality and HDL-C or LDL-C concentrations in the subgroup of patients with bacterial respiratory coinfection supports this hypothesis. However, these preliminary results suggest that the viral inflammatory model has a major impact on lipoprotein metabolism and deserves additional investigation. In particular, the dysfunction of lipoproteins (and especially of HDL particles) may be interesting to explore in relation to COVID-19 pneumonia [45, 46]. Because no patients in our cohort had bacterial coinfection at admission and owing to the late delay before the onset of VAP (7 [3–9] days), COVID-19 appears to be an interesting model to 1) study the effects of viral infection and 2) the effect of bacterial VAP on the lipid profile.

Interestingly, compared to the HDL-C, LDL-C and TC concentrations, the TG concentration at admission was higher than the normal range. Several hypotheses can explain this finding. First, one-third of the patients had diabetes, a condition in which lipid metabolism disorders are frequent (particularly increased TG concentrations). Second, the early use of propofol upon admission may explain the increased TG concentration. Third, renal dysfunction and especially nephrotic-like syndrome is observed in COVID-19 disease, which may have induced an increased level of TG. Finally, direct effects of the virus cannot be excluded.

Our study has several limitations. First, it is a small monocentric study of limited size. Second, we did not measure cytokine levels; these could have been used to stratify the patients according to their inflammatory state. Third, we did not compare lipid concentrations upon admission with basal concentrations prior to hospitalization. Fourth, although the lipid concentrations of patients treated with and without corticosteroids or protease inhibitors were

similar over time, these treatments have the potential to induce lipids disorders (such as increasing the TC, TG, LDL-C concentrations and decreasing the HDL-C concentration), which can influence patient outcomes. Finally, patients with diabetes mellitus had a higher mortality, which may have introduced bias. Diabetes itself affects lipid levels (e.g., decreases HDL-C levels and increases TG levels) and could have played a much more important role in death than did the altered lipid levels.

## Conclusion

HDL-C and LDL-C concentrations were low in patients upon ICU admission for severe COVID-19, but they were not associated with poor outcomes. The low lipoprotein concentrations observed in bacterial superinfections were associated with mortality. These findings reinforce the hypothesis that lipoproteins play a role in bacterial sepsis. Studies with higher statistical power are needed to better characterize the role of lipoproteins during severe COVID-19. Further experimental studies evaluating the functionality of these particles, in particular HDL, are crucial to understand the physiopathology of this disease.

## Supporting information

**S1 Fig. Comparison of the total cholesterol, triglycerides, HDL-C and LDL-C concentration kinetics over the first six days in patients with and without lopinavir/ritonavir treatment.**
(TIF)

**S2 Fig. Comparison of the total cholesterol, triglycerides, HDL-C and LDL-C concentration kinetics over the first six days in patients with and without corticosteroid treatment.**
(TIF)

**S1 Table. Correlation between TC, TG, HDL-C and LDL-C concentrations and the number of days alive without mechanical ventilation, ICU length of stay, and hospital length of stay.**
(TIF)

## Acknowledgments

We thank Pr Romain SONNEVILLE (Department of Intensive Care Medicine and Infectious Diseases, APHP, Bichat-Claude Bernard Hospital, Paris, France) for his scientific advice.

## Author Contributions

**Conceptualization:** Sébastien Tanaka, Christian De Tymowski, Tiphaine Robert, Katell Peoc'h, Lauriane Fontaine, Alexy Tran-Dinh, Olivier Meilhac, Philippe Montravers.

**Data curation:** Sébastien Tanaka, Sylvain Jean-Baptiste, Tiphaine Robert, Katell Peoc'h, Brice Lortat-Jacob, Donia Bouzid, Alexy Tran-Dinh, Parvine Tashk.

**Formal analysis:** Sébastien Tanaka, Christian De Tymowski, Maksud Assadi, Nathalie Zappella, Brice Lortat-Jacob, Lauriane Fontaine, Parvine Tashk, Philippe Montravers.

**Investigation:** Sébastien Tanaka, Christian De Tymowski, Maksud Assadi, Nathalie Zappella, Sylvain Jean-Baptiste, Tiphaine Robert, Katell Peoc'h, Brice Lortat-Jacob, Donia Bouzid, Alexy Tran-Dinh, Parvine Tashk, Olivier Meilhac, Philippe Montravers.

**Methodology:** Sébastien Tanaka, Christian De Tymowski, Alexy Tran-Dinh, Olivier Meilhac, Philippe Montravers.

**Resources:** Sébastien Tanaka, Olivier Meilhac.

**Supervision:** Sébastien Tanaka, Olivier Meilhac, Philippe Montravers.

**Validation:** Sébastien Tanaka, Olivier Meilhac, Philippe Montravers.

**Visualization:** Olivier Meilhac, Philippe Montravers.

**Writing – original draft:** Sébastien Tanaka, Olivier Meilhac, Philippe Montravers.

**Writing – review & editing:** Sébastien Tanaka, Christian De Tymowski, Maksud Assadi, Nathalie Zappella, Sylvain Jean-Baptiste, Tiphaine Robert, Katell Peoc'h, Brice Lortat-Jacob, Lauriane Fontaine, Donia Bouzid, Alexy Tran-Dinh, Parvine Tashk, Olivier Meilhac, Philippe Montravers.

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
