## [Decision Letter · Decision Letter 0]

14 Aug 2020

PONE-D-20-20848

Lipoprotein concentrations over time in the intensive care unit COVID-19 patients:

Results from the ApoCOVID study

PLOS ONE

Dear Dr. TANAKA,

Thank you for submitting your manuscript to PLOS ONE. After careful consideration, we feel that it has merit but does not fully meet PLOS ONE’s publication criteria as it currently stands. Therefore, we invite you to submit a revised version of the manuscript that addresses the points raised during the review process.

We look forward to receiving your revised manuscript.

Kind regards,

Wenbin Tan

Academic Editor

PLOS ONE

Comments from the Editor:

1. The authors need to specify that the lipid profiles were determined at the time of ICU admission in the abstract, figure legends and figures. For example in the conclusion of abstract, "HDL-c and LDL-c concentrations at the ICU admission are..". and many other places in the contents. 

2. justification of using LDL <1.4 mM as the cut-off value needs to be elaborated?

3. how many patients died after 28 days? what is the justification of using 28 days at ICU? whether it can add the lipid profiles at time of discharge or death for analyses instead of 28 days?

4. Had the patients stayed in non-ICU ward before being admitted into ICU? if so, did the authors have the lipid profiles of these patients in this cohort on the non-ICU ward admission? Would the LDL or HDL levels be higher than on the time of ICU admission for those survival patients?

5. the authors need to cite references *Clinica chimica acta; *2020:10.1016/j.cca.2020.07.015. *and FASEB journal *2020:10.1096/fj.202001451.

6. the authors need to explain the possible reason why other reports (refs 28, 32, and 2020:10.1016/j.cca.2020.07.015)  showed the association with disease severity because that they used lipid profiles at patient hospitalization admission which levels were generally higher than at the time of ICU admission for those surviving patients.

2. Please provide additional details regarding participant consent. In the ethics statement in the Methods and online submission information, please ensure that you have specified (1) whether consent was informed and (2) what type you obtained (for instance, written or verbal, and if verbal, how it was documented and witnessed). If the need for consent was waived by the ethics committee, please include this information.

Review Comments to the Author

Reviewer #1: The goal of the study was to characterize the lipoprotein profiles of severe ICU patients hospitalized for COVID-19 pneumonia and to assess their changes during bacterial ventilator-associated pneumonia (VAP) superinfection.

The authors found that HDL-C and LDL-C concentrations are low during severe COVID-19 pneumonia but are not associated with poor outcomes, and that low lipoproteins in the case of bacterial superinfection during ICU hospitalization are associated with mortality.

Comments

1. The presence of diabetes or hyperglycemia is associated with worse prognosis of severe Covid-19 pneumonia. How was a difference in glycemic control during hospitalization between alive patients and dead patients at 28 days?

2. Did you look at changes in inflammatory markers, including leukocytosis and C-reactive protein 6, during hospitalization between alive patients and dead patients at 28 days?

3. Cytokine storm contributes to poor outcome of patients with severe COVID-19. Did you look at changes in cytokine levels (IL-2R, IL-6, TNF-alpha) during hospitalization?

4. Did the use of statin affect prognosis of severe Covid-19 pneumonia?

5. Please add prevalence of use of ACEI or ARB to Table 1.

Reviewer #2:

The manuscript definitely needs English language editing.The authors use SARS-CoV-2 interchangeably with COVID-19. However, being a clinical study (i.e., not basic science), it is advisable if they stick to using the name of the disease i.e., COVID-19, and not the name of the virus (SARS-CoV-2) which is to be reserved mostly for mechanistic and preclinical studies.In the abstract they mention “the short-term prognosis outcome was assessed”. That is a very vague statement and requires clarification by saying that survival at 28 days post admission with incidence of VAP was assessed.They mention that LDL-C levels have been previously correlated with some viral infections (eg, Dengue fever) but do not clarify the direction of the correlation (direct/positive or inverse/negative)?Table 1 shows that diabetes was far more prevalent among the patients who died at the end of the study so that is a very major confounding factor that the authors fail to discuss. Diabetes itself affects lipid levels, and could have played a much more important role in death than did the altered lipid levels. This should be addressed.Also, the use of corticosteroids an protease inhibitors differed between those who were alive vs dead at the end of the study (even if not significant; it was a marked difference in rates). Both drugs are associated with marked variation and have different effects on lipoprotein levels so these too are major confounders that should be noted and discussed as the differences in lipoprotein levels between the two groups could simply be a result of the different rates of using these drugs among the groups.All patients had higher than normal TG levels at admission which also warrants discussion.They also fail to discuss the direction of associations found in their studies or other similar studies they cite. They do not mention whether the associations were positive or negative, which is crucial to clarify throughout the manuscript, including when citing other studies. Also, refrain from saying “statistical association” and replace with “statistically significant association”.Finally, since many COVID-19 patients develop a nephrotic-like syndrome, that can be the reason for alteration of the lipid levels and not the infection per se. This should be noted and clearly discussed in the manuscript.

---

## [Author Response · Author response to Decision Letter 0]

2 Sep 2020

Dear Editor, 

Thank you for reviewing our manuscript entitled “Lipoprotein concentrations over time in the intensive care unit COVID-19 patients: Results from the ApoCOVID study”. We appreciate your positive comments and relevant questions that will help to improve the manuscript. Please find attached a revised version of our manuscript and a point-by-point response to the editor and reviewer comments. 

Comments from the Editor:

1. The authors need to specify that the lipid profiles were determined at the time of ICU admission in the abstract, figure legends and figures. For example in the conclusion of abstract, "HDL-c and LDL-c concentrations at the ICU admission are..". and many other places in the contents. 

In accordance with this comment, we have added the term “at ICU admission” in the abstract, manuscript, figure legends and figures. 

2. justification of using LDL <1.4 mM as the cut-off value needs to be elaborated?

To evaluate the impact of low LDL-C values on outcomes, the first step was to determine the threshold of a low LDL-C level. As no cutoff is consensual, our population was split into four quartiles according to LDL-C level, and the threshold to determine the lowest quartile was retained as the cutoff value. This value was 1.4 mM. 

This point is mentioned in the statistics section: ”The threshold defining the lower quartile was determined to have 25% of the overall population in that quartile”.

3. how many patients died after 28 days? what is the justification of using 28 days at ICU? whether it can add the lipid profiles at time of discharge or death for analyses instead of 28 days?

Thank you for raising this important question. Only one patient died after 28 days. He died at day 67. A 28-day endpoint is often applied in studies involving ICU patients, especially for sepsis because of the high mortality rate of this condition, particularly in the first month. In this context, although this cutoff has become more controversial according to some authors, a 28-day mortality endpoint is relevant (Vincent et al. Crit Care Med 2004 PMID 15118519).

We collected all lipid concentrations from ICU admission to discharge or death. To perform the analyses, we had to choose a mortality endpoint. In most cases, the death of our ICU patients occurred during the first month (median delay = 7 days [7-13]). In this context, and regarding the literature in the field, we selected the 28-day endpoint. 

4. Had the patients stayed in non-ICU ward before being admitted into ICU? if so, did the authors have the lipid profiles of these patients in this cohort on the non-ICU ward admission? Would the LDL or HDL levels be higher than on the time of ICU admission for those survival patients?

A total of 29 patients were directly admitted to the ICU, and 19 patients stayed in the non-ICU ward before being admitted to the ICU. The median delay between hospital and ICU admission was very short (1 [0-3] days), and unfortunately lipid profiles of non-ICU patients are not routinely performed. 

We added the number of patients who were directly admitted to ICU in the Results section: “Twenty-nine patients were directly admitted to the ICU, while 19 patients had a short stay (1 [0-3] days) in another ward before ICU admission.”

5. the authors need to cite references Clinica chimica acta; 2020:10.1016/j.cca.2020.07.015. and FASEB journal 2020:10.1096/fj.202001451.

In accordance with this suggestion, we have added these interesting references to the Discussion section. 

6. the authors need to explain the possible reason why other reports (refs 28, 32, and 2020:10.1016/j.cca.2020.07.015) showed the association with disease severity because that they used lipid profiles at patient hospitalization admission which levels were generally higher than at the time of ICU admission for those surviving patients.

Thank you for this relevant question. 

Fan et al. (Fan et al. Metabolism 2020 PMID 32320740) collected lipid profiles of 21 COVID-19 patients. Compared to data collected before falling ill with COVID-19 pneumonia, the authors showed that TC and LDL-C levels were decreased at the time of admission and then returned to the level prior to infection. The HDL-C level also decreased but remained low over time. Interestingly, logistic regression analysis showed increasing odds of a lowered LDL level being associated with disease progression and in-hospital mortality. 

In the Wei et al. study (Wei et al. J of Clinical Lipidology 2020 PMID 32430154) involving 597 COVID-19 patients and control patients, the authors showed that COVID-19 patients had lower LDL-C, HDL-C and TG levels at admission compared with control patients. When patients were stratified as mild/severe and critical, the HDL-C levels were lower in the critical patients than in the mild/severe patients. 

Hu et al. (Hu et al. Clinica Chimica Acta PMID 32653486) analyzed 114 COVID-19 cases and 80 age-matched healthy controls. TC, HDL-C, LDL-C levels were significantly decreased in COVID-19 patients compared to healthy controls. Additionally, HDL-C levels were lower in the severe group than in the common group. The authors concluded that decreased serum HDL-C was associated with an increased severity of COVID-19.

These different studies demonstrate that hypolipidemia, and in particular a decreased HDL-C concentration, is associated with severe disease. Interestingly, high variability was observed in patient severity in these studies, with very few patients hospitalized in an ICU, while the majority was hospitalized in a non-ICU ward. This allows stratification of the patients according to severity. Since our patients were all severe ICU patients, stratification was not performed. In addition, the results of these studies cannot be directly compared with those of our work. It should also be noted that these studies compared data from COVID-19 patients with healthy volunteers. We did not perform this type of comparison.

We further addressed this comment in the Discussion section: “In our cohort of 48 severe COVID-19 patients hospitalized in our ICU, we found no statistically significant link between HDL-C or LDL-C concentrations and patient outcomes. These results are not in accordance with the results of the study by Fan et al., which showed that a low LDL-C concentration was a potential predictor of poor prognosis in a cohort of 21 heterogeneous ICU and non-ICU COVID-19 patients (30). Interestingly, the studies by Hu et al. and Wei et al. have specifically shown that low HDL-C levels were associated with severe COVID-19 disease (34,39). However, the heterogeneity reported in these two studies, which included mild, severe and critical patients and ICU and non-ICU patients, does not allow for direct comparisons with our specific severe COVID-19 ICU cohort (34,39)”.

https://clicktime.symantec.com/3187EePMb4iL9YFXPcRHTy26H2?u=https%3A%2F%2Fjournals.plos.org%2Fplosone%2Fs%2Ffile%3Fid%3DwjVg%2FPLOSOne_formatting_sample_main_body.pdf and

https://clicktime.symantec.com/3RE1YxDqS7h8kUXmYktK8Ap6H2?u=https%3A%2F%2Fjournals.plos.org%2Fplosone%2Fs%2Ffile%3Fid%3Dba62%2FPLOSOne_formatting_sample_title_authors_affiliations.pdf

 The manuscript was modified according to PLOS One’s style requirements. 

2. Please provide additional details regarding participant consent. In the ethics statement in the Methods and online submission information, please ensure that you have specified (1) whether consent was informed and (2) what type you obtained (for instance, written or verbal, and if verbal, how it was documented and witnessed). If the need for consent was waived by the ethics committee, please include this information.

The need for consent was waived by the ethics committee. We have added this information in the Methods section and in the online submission information section. 

3. PLOS requires an ORCID iD for the corresponding author in Editorial Manager on papers submitted after December 6th, 2016. Please ensure that you have an ORCID iD and that it is validated in Editorial Manager. To do this, go to ‘Update my Information’ (in the upper left-hand corner of the main menu), and click on the Fetch/Validate link next to the ORCID field. This will take you to the ORCID site and allow you to create a new iD or authenticate a pre-existing iD in Editorial Manager. Please see the following video for instructions on linking an ORCID iD to your Editorial Manager account: https://clicktime.symantec.com/393EgirqgVZPnoGeJag8CU46H2?u=https%3A%2F%2Fwww.youtube.com%2Fwatch%3Fv%3D_xcclfuvtxQ

 We have added the ORCID iD online, which is validated in the Editorial Manager. 

Review Comments to the Author

Reviewer #1: The goal of the study was to characterize the lipoprotein profiles of severe ICU patients hospitalized for COVID-19 pneumonia and to assess their changes during bacterial ventilator-associated pneumonia (VAP) superinfection.

The authors found that HDL-C and LDL-C concentrations are low during severe COVID-19 pneumonia but are not associated with poor outcomes, and that low lipoproteins in the case of bacterial superinfection during ICU hospitalization are associated with mortality.

Comments

1. The presence of diabetes or hyperglycemia is associated with worse prognosis of severe Covid-19 pneumonia. How was a difference in glycemic control during hospitalization between alive patients and dead patients at 28 days?

As reported in numerous studies, we found that diabetes mellitus was associated with a worse prognosis (mortality at 28 days in diabetic patients = 56% vs. 25% in non-diabetic patients, p = 0.033).

In our ICU, glycemic control during hospitalization was standardized for all patients (diabetic and non-diabetic patients) according to the most recent recommendations (Surviving Sepsis Campaign International Guidelines for Management of Sepsis and Septic Shock, Critical Care Medicine 2016). We used a glycemic control protocol. If two consecutive blood glucose levels were >180 mg/dl, IV or subcutaneous insulin was administered. The target was an upper blood glucose level ≤180 mg/dl. Blood glucose values were monitored every 1 to 2 hours until the glucose values and insulin infusion rates were stable, then every 4 hours thereafter in patients receiving insulin infusion. 

With this standardized glycemic control protocol, we found no difference in glycemic control during hospitalization between survivors and nonsurvivors at 28 days. 

2. Did you look at changes in inflammatory markers, including leukocytosis and C-reactive protein 6, during hospitalization between alive patients and dead patients at 28 days? 

We thank the reviewer for this interesting question. 

At admission, no significant difference was noted between survivors and nonsurvivors at 28 days in the levels of leukocytes, lymphocytes, C-reactive protein and procalcitonin. Furthermore, no significant differences were noted during the course of ICU stay, but survivors tended to have higher lymphocyte counts. This observation is in accordance with several studies reporting that lymphopenia leads to poor outcomes in COVID-19 patients (Li tan et al. signal transduction and targeted therapy, 2020, lymphopenia predicts disease severity of COVID-19, PMID 32296069; Zhao et al. International journal of infectious diseases, lymphopenia is associated with severe coronavirus disease, a systematic review and meta-analysis, PMID 32376308).

We added this information (leukocytes, lymphocytes, C-reactive protein and procalcitonin values) to Table 1. 

3. Cytokine storm contributes to poor outcome of patients with severe COVID-19. Did you look at changes in cytokine levels (IL-2R, IL-6, TNF-alpha) during hospitalization?

Unfortunately, we did not monitor cytokine levels in our study. We added this point to the limitations of the study. 

4. Did the use of statin affect prognosis of severe Covid-19 pneumonia?

We thank the reviewer for this question. A recent retrospective study involving 1,219 patients receiving statins and using propensity score-matching showed that the risk for 28-day all-cause mortality was 5.2% and 9.4%, respectively, in the matched statin and non-statin groups, with an adjusted hazard ratio of 0.58 (Zhang et al. Cell Metab. 2020 PMID 32592657). Other studies did not find this association. In our cohort of severe COVID-19 patients, as indicated in Table 1, statin medication did not affect prognosis (22% in survivors vs. 37% in nonsurvivors, p = 0.310). 

5. Please add prevalence of use of ACEI or ARB to Table 1.

We found no difference between survivors and nonsurvivors. We have added this information to Table 1.

Reviewer #2:

1. The manuscript definitely needs English language editing.

We used an English language editing service to improve the quality of the manuscript (AJE certificate number D83B-0D4E-75F4-9249-C616, September 1,2020).

2. The authors use SARS-CoV-2 interchangeably with COVID-19. However, being a clinical study (i.e., not basic science), it is advisable if they stick to using the name of the disease i.e., COVID-19, and not the name of the virus (SARS-CoV-2) which is to be reserved mostly for mechanistic and preclinical studies.

Thank you for this relevant comment. We changed the term SARS-CoV-2 to COVID-19 throughout the manuscript.

3. In the abstract they mention “the short-term prognosis outcome was assessed”. That is a very vague statement and requires clarification by saying that survival at 28 days post admission with incidence of VAP was assessed.

We changed “the short-term prognosis outcome was assessed” to “The assessed outcomes were survival at 28 days and the incidence of VAP”.

4. They mention that LDL-C levels have been previously correlated with some viral infections (eg, Dengue fever) but do not clarify the direction of the correlation (direct/positive or inverse/negative)?

We apologize for this inconsistency. We clarified the direction of the correlations in the manuscript. 

- Marin-Palma et al. showed that dengue patients had significantly lower HDL-C, LDL-C and TC levels and higher triglyceride level compared to healthy controls (Marin-Palma et al. Plosone. 2019 PMID 30901375). Interestingly, as described by Biswa et al. in patients with dengue disease, lower LDL-C levels were associated with severe dengue outcomes (Biswa et al. plos neg trop dis 2015 PMID 26334914). We have added these modifications and references to the Introduction section. 

- Interestingly, lipid profiles depend on the progression of hepatitis B disease. Cao et al. found that TC, HDL-C and LDL-C are mostly low in cases of hepatitis B-related cirrhosis (Cao et al. Clin lab 2020 PMID 31850701). 

- HIV infection and its treatment are frequently associated with dyslipidemia, with elevated TC and LDL-C levels and decreased HDL-C levels (Rose et al. Atherosclerosis. 2008 PMID 18054941). We added this point to the Discussion section. 

5. Table 1 shows that diabetes was far more prevalent among the patients who died at the end of the study so that is a very major confounding factor that the authors fail to discuss. Diabetes itself affects lipid levels, and could have played a much more important role in death than did the altered lipid levels. This should be addressed.

Thank you for this important point to discuss. As reported in numerous studies, we also observed a worse prognosis in diabetic patients (mortality at 28 days in diabetic patients = 56% vs. 25% in non-diabetic patients, p = 0.033). The cause of this excess mortality is not well-described for the population of diabetic COVID-19 patients. Diabetic patients frequently have dyslipidemia and, in particular, decreased HDL-C and increased triglycerides. 

As proposed by the reviewer, diabetes itself could have played a much more important role in death than did the altered lipid levels. We agree with the reviewer all the more because we found no relationship between lipoprotein levels are mortality in our cohort. We remarked upon this in the limitations paragraph in the Discussion section. 

6. Also, the use of corticosteroids an protease inhibitors differed between those who were alive vs dead at the end of the study (even if not significant; it was a marked difference in rates). Both drugs are associated with marked variation and have different effects on lipoprotein levels so these too are major confounders that should be noted and discussed as the differences in lipoprotein levels between the two groups could simply be a result of the different rates of using these drugs among the groups.

Thank you for raising this interesting point. Indeed, patients who received protease inhibitors had a lower mortality rate (17%) than the overall mortality rate of 33%. 

Protease inhibitors can induce chronic dyslipidemia; this has primarily been described in HIV patients who have increased TC, TG, and LDL-C levels and a variable decrease in HDL-C levels (Berthold et al. Journal of Internal Medicine 1999 PMID 10620100, Roberts et al. CID 1999 PMID 10476757, Montes et al. JAC. 2005 PMID 15761071). Thanks to the reviewer, we compared lipid levels between patients with and without protease inhibitor treatment and found no differences over time (see new supplemental figure S1). This may be due to the fact that the altered lipid concentrations appear long time after the initiation of therapy (for example, in the study by Roberts et al, lipid profile alterations were observed 33 months after treatment initiation). In our study, the treatment duration was likely too short to see an impact of lopinavir/ritonavir on lipid metabolism.

However, undeniably, protease inhibitors could induce a bias; therefore, we included this point in the limitations. 

Even more controversial than protease inhibitors, corticosteroids can also induce chronic changes in lipid levels, such as increased levels of TC, TG and LDL-C. Similarly, no differences were noted between patients with or without corticosteroid treatment (see new supplemental figure S2). This point was also added to the Discussion. 

7. All patients had higher than normal TG levels at admission which also warrants discussion.

Thank you for this interesting point. We added this issue to the discussion: “Interestingly, compared to the HDL-C, LDL-C and TC concentrations, the TG concentration at admission was higher than the normal range. Several hypotheses can explain this finding. First, one-third of the patients had diabetes, a condition in which lipid metabolism disorders are frequent (particularly increased TG concentrations). Second, the early use of propofol upon admission may explain the increased TG concentration. Third, renal dysfunction and especially nephrotic-like syndrome is observed in COVID-19 disease, which may have induced an increased level of TG. Finally, direct effects of the virus cannot be excluded.”

8. They also fail to discuss the direction of associations found in their studies or other similar studies they cite. They do not mention whether the associations were positive or negative, which is crucial to clarify throughout the manuscript, including when citing other studies. 

Thank you for this comment. We clarified the purpose in the Discussion section of the manuscript.

Also, refrain from saying “statistical association” and replace with “statistically significant association”.

We changed the term “statistical association” to “statistically significant association”.

9. Finally, since many COVID-19 patients develop a nephrotic-like syndrome, that can be the reason for alteration of the lipid levels and not the infection per se. This should be noted and clearly discussed in the manuscript.

Nephrotic syndrome induces lipid disorders with increased concentrations of TC and TG. Although we did not note nephrotic-like syndrome in our cohort of COVID-19 patients, this syndrome has been described in COVID-19 pneumonia and could explain part of the altered lipid levels, especially the increased TG concentrations, in our patient cohort. We added this point to the Discussion section of the manuscript: “Third, renal dysfunction and especially nephrotic-like syndrome is observed in COVID-19 disease, which may have induced an increased level of TG.”

---

## [Decision Letter · Decision Letter 1]

10 Sep 2020

Lipoprotein concentrations over time in the intensive care unit COVID-19 patients:

Results from the ApoCOVID study

PONE-D-20-20848R1

Dear Dr. TANAKA,

We’re pleased to inform you that your manuscript has been judged scientifically suitable for publication and will be formally accepted for publication once it meets all outstanding technical requirements.

Kind regards,

Wenbin Tan

Academic Editor

PLOS ONE

Additional Editor Comments:

Please consider to add the following reference into the content during your proofreading:

https://doi.org/10.1038/s41392-020-00292-7

Reviewers' comments:

Reviewer #1: This manuscript has been considerably improved. I feel no significant concerns about this revised manucsript.

---

## [Editor Report · Acceptance letter]

17 Sep 2020

PONE-D-20-20848R1 

Lipoprotein concentrations over time in the intensive care unit
COVID-19 patients: Results from the ApoCOVID study 

Dear Dr. TANAKA:

I'm pleased to inform you that your manuscript has been deemed suitable for publication in PLOS ONE. Congratulations! Your manuscript is now with our production department. 

Kind regards, 

on behalf of

Dr. Wenbin Tan 

Academic Editor

PLOS ONE